Trace elements during primordial plexiform network formation in human cerebral organoids

Sartore Rafaela C. 1 2
Cardoso Simone C. 3
Lages Yury V.M. 1 2
Paraguassu Julia M. 1 2
Stelling Mariana P. 4
Madeiro da Costa Rodrigo F. 1
Guimaraes Marilia Z. 1 2
Pérez Carlos A. 5
Rehen Stevens K. srehen@lance-ufrj.org 1 2
1 D’Or Institute for Research and Education (IDOR) , Rio de Janeiro , Brazil
2 Institute of Biomedical Sciences, Federal University of Rio de Janeiro , Brazil
3 Physics Institute, Federal University of Rio de Janeiro , Brazil
4 Federal Institute of Education, Science and Technology of Rio de Janeiro , Brazil
5 Brazilian Synchrotron Light Laboratory , São Paulo , Brazil
Reser David
Electronic publication date: 2017 Feb 8
Publication date: 2017
Volume: 5
Electronic Location ID: e2927
Received 2016 Jul 25; Accepted 2016 Dec 20
Copyright: ©2017 Sartore et al.
Copyright year: 2017
Copyright holder: Sartore et al.
License: This is an open access article distributed under the terms of the Creative Commons Attribution License, which permits unrestricted use, distribution, reproduction and adaptation in any medium and for any purpose provided that it is properly attributed. For attribution, the original author(s), title, publication source (PeerJ) and either DOI or URL of the article must be cited.
License URL: https://creativecommons.org/licenses/by/4.0/

Keywords: Trace elements, Cerebral organoids, Development, SR-XRF, Neurogenesis, Human pluripotent stem cells

Funding: National Council for Scientific and Technological Development (CNPq) Foundation for Research Support in the State of Rio de Janeiro (FAPERJ) Coordenação de Aperfeiçoamento de Pessoal de Nível Superior (CAPES) Funding Authority for Studies and Projects (FINEP) Brazilian Development Bank (BNDES) Laboratório Nacional de Luz Síncroton This study was funded by the following Brazilian funding agencies: National Council for Scientific and Technological Development (CNPq), Foundation for Research Support in the State of Rio de Janeiro (FAPERJ), Coordenação de Aperfeiçoamento de Pessoal de Nível Superior (CAPES), Funding Authority for Studies and Projects (FINEP), Brazilian Development Bank (BNDES) and Laboratório Nacional de Luz Síncroton. The funders had no role in study design, data collection and analysis, decision to publish, or preparation of the manuscript.

==============================
Systematic studies of micronutrients during brain formation are hindered by restrictions to animal models and adult post-mortem tissues. Recently, advances in stem cell biology have enabled recapitulation of the early stages of human telencephalon development in vitro. In the present work, we analyzed cerebral organoids derived from human pluripotent stem cells by synchrotron radiation X-ray fluorescence in order to measure biologically valuable micronutrients incorporated and distributed into the exogenously developing brain. Our findings indicate that elemental inclusion in organoids is consistent with human brain tissue and involves P, S, K, Ca, Fe and Zn. Occurrence of different concentration gradients also suggests active regulation of elemental transmembrane transport. Finally, the analysis of pairs of elements shows interesting elemental interaction patterns that change from 30 to 45 days of development, suggesting short- or long-term associations, such as storage in similar compartments or relevance for time-dependent biological processes. These findings shed light on which trace elements are important during human brain development and will support studies aimed to unravel the consequences of disrupted metal homeostasis for neurodevelopmental diseases, including those manifested in adulthood.

Introduction

Cerebral development is a lifelong event beginning almost immediately after fertilization. During the third embryonic week, neural tube is formed and before the end of the following week, it compartmentalizes into forebrain, midbrain, and hindbrain. Beginning from the forebrain, the telencephalon evolves towards a complex network with billions of neurons and glial cells in the cerebral cortex, from which organized human thought and behavior will emerge. To a large extent, the blueprint for the postnatal brain is laid out during gestation, when fetal neural cells proliferate, differentiate, migrate, and make connections with other cells. These activities, for the most part, appear to be genetically determined, epigenetically directed, and influenced by the physical and chemical environment of the womb (Georgieff, 2007; Paridaen & Huttner, 2014). Hence, the healthy development of central nervous system (CNS) in the period between the third and seventh week of embryo development is likely to rely on adequate provisions of maternal resources like vitamins or dietary elements.

Nutritional resources available to an embryo in these earliest weeks of life putatively determine some of the most important aspects of future health. One well understood example that relates maternal diet to CNS pathologies is spina bifida, caused by insufficient folic acid in early pregnancy. While less understood, abnormalities in the levels of essential minerals potentially begin in utero (Radlowski & Johnson, 2013; Golub & Hogrefe, 2015). Examples of neurological disorders in which micronutrient imbalances have been identified in adulthood include Alzheimer’s, Parkinson’s, and Huntington’s diseases (Miller et al., 2006; Popescu et al., 2009; Rosas et al., 2012). It is, however, difficult to assess dynamic trace element changes in the developing human brain. Determining expected mineral levels according to developmental stage are of clear importance; however, few studies have successfully addressed this issue. To date, the understanding of mineral inclusion as part of human brain development has only been carried out in post-mortem tissue often involving fixative-treated brains, and inferred from animal models (Wróblewski, Chamberlain & Edström, 1984; Rajan et al., 1997). While analyses on this subject have been conducted on a variety of species, numerous peculiarities including cell types and a distinctive temporal organization make the human brain unique and substantiate the need for in-depth studies of these phenomena in human tissue. One method recently refined to model cellular and molecular events of human embryonic brain development is growing cerebral organoids in vitro (Eiraku et al., 2008; Lancaster et al., 2013). These three-dimensional structures, derived from human pluripotent stem cells, progressively differentiate and self-organize into physiologically relevant cellular niches that mirror the developing human brain.

In the present work, we used synchrotron radiation based micro X-ray fluorescence (SR-XRF) analysis to detect and quantify trace elements present in human cerebral organoids. We sought to capture the levels and the distribution of minerals in brain tissue during a period of intense cell proliferation versus one in which early neuronal network formation was a dominant developmental feature. This work is the first description of chemical elements composition and distribution in human cerebral organoids.

Materials & Methods

Generation of human induced pluripotent stem cells

Human induced pluripotent stem (iPS) cells were obtained from a skin biopsy as described previously (Sochacki et al., 2016). Briefly, fibroblasts were maintained in culture and then transduced with CytoTune-iPS Sendai reprogramming kit 2.0 (Thermo-Fischer) as per manufacturer’s instructions. Then, following colony formation and expansion, cells were checked for pluripotency markers via RT-PCR and ability to differentiate into embryoid bodies (Fig. S1). Skin tissue was obtained after donor signed an informed consent approved by the Research Ethics Committee of Hospital das Clínicas de Porto Alegre (CAPPesq, HCPA, IRB00000921) and by the Research Ethics Committee of Hospital Copa D’Or Rio de Janeiro (CEPCOPADOR, number 727.269).

Human pluripotent stem cells

Human embryonic stem cells (hESC, BR1 cell line) (Fraga et al., 2011) and iPS cells were cultured in mTSeR1 medium (Stemcell Technologies) on Matrigel-coated surface (BD Biosciences). The colonies were manually passaged every seven days and maintained at 37 °C in humidified air with 5% CO2.

Human cerebral organoids

Pluripotent cell differentiation into cerebral organoids was based in a previously described protocol (Lancaster et al., 2013). However, our protocol was conducted mostly using spinner flasks under continuous rotation. Briefly, human pluripotent stem cells were dissociated with Accutase (Millipore) until obtainment of a single-cell solution. Then, approximately 250,000 cells/mL were inoculated into a spinner flask containing mTeSR1 to final volume of 50 mL, supplemented with 10 µM Y-27632 (Rho-associated protein kinases inhibitor, iRock) (Merck, Millipore) under uninterrupted rotation (40 rpm). After 24 h, medium was changed to Dulbecco’s modified eagle medium (DMEM)/F12, supplemented with 20% KnockOut™ Serum Replacement (KOSR, Invitrogen), 2 mM Glutamax (Invitrogen), 1% minimum essential medium nonessential amino acids (MEM-NEAA, Gibco), 55 µM 2-Mercaptoethanol (Gibco) and 100 U/mL Penicillin-Streptomycin (Gibco). By day 7, embryoid bodies (EB) were fed with neuroinduction medium composed of DMEM/F12, 1× N2 supplement (Gibco), 2 mM Glutamax (Invitrogen), 1% MEM-NEAA and 1 µg/mL heparin (Sigma) for four days. On day 11, cellular aggregates were transferred to petri dishes and embedded in Matrigel for 1 h at 37 °C and 5% CO2. Then, cellular aggregates were decanted in a conical tube and returned to a spinner flask containing neurodifferentiation medium composed of 1:1 DMEM/F12: Neurobasal (Gibco), 0.5x N2, 1x B27 minus vitamin A (Gibco), 2 mM Glutamax, 0.5% MEM-NEAA, 0.2 µM 2-Mercaptoethanol and 2.5 µg/mL insulin. After 4 days, cellular aggregates were grown in the aforementioned medium except by replacing with B27 containing vitamin A (Gibco). The medium was changed every week. Cerebral organoids were grown until 30 days of differentiation (totalizing 15 days in neurodifferentiation medium containing vitamin A) and 45 days (totalizing 30 days in neurodifferentiation medium containing vitamin A) for analyses. The cerebral organoids derived from embryonic stem cells were obtained from two independent assays.

Measurements of cerebral organoid diameter and measurements of epithelium-lined cavities total area and number

Cerebral organoids were transferred to non-adherent petri dishes and photographed with an inverted microscope (Eclipse TS100, Nikon). Using ImageJ software (Rasband, W.S., ImageJ, U. S. National Institutes of Health, Bethesda, Maryland, USA, http://imagej.nih.gov/ij/, 1997–2016), cerebral organoid major diameter was measured using the straight-line tool, with reference to the scale bar. The analyzed number of cerebral organoids derived from hESC was as follows: 7-days old organoids, n = 107; 15-days old organoids, n = 90; 30-days old organoids, n = 56; 45-days old organoids, n = 18, obtained from two independent experiments. For cerebral organoids derived from iPS cells, the number of cerebral organoids analyzed was as follows: 7-days old organoids, n = 8; 15-days old organoids, n = 15; 30-days old organoids, n = 7; 45-days old organoids, n = 10, obtained from one experiment.

To quantify the number of ventricle-like epithelium-lined cavities and to measure their luminal area, tissue sections were stained with hematoxylin and eosin (H&E). Analyzed structures were only considered as ventricle-like cavities when surrounded by a stratified epithelium containing radially organized cells. The number of independent organoids inspected to establish the number of epithelium-lined cavities per organoid was: 30-days old organoids, n = 6; 45-days old organoids, n = 11. The luminal area was measured with ImageJ software, through delimitation with the freehand selection tool. The number of analyzed ventricles was: 30-days old organoids, n = 17 sections from six independent cerebral organoids; 45-days old organoids, n = 9 sections from 11 independent cerebral organoids.

Immunohistochemistry

Cerebral organoids were fixed in 4% paraformaldehyde, sequentially incubated in sucrose solutions (10, 20 and 30%) prepared in phosphate buffered saline (PBS), embedded in optimal cutting temperature compound (OCT) and frozen in liquid nitrogen. The organoids were sectioned with a cryostat (Leica) into 20 µm thick sections. Immunofluorescence was performed using the primary antibodies: anti-Nestin (MAB5326; Chemicon), anti-PAX6 (sc11357; Santa Cruz), anti-TBR2 (AB2283; Millipore), anti-class III β-tubulin (T3952; Sigma Aldrich), anti-MAP2 (M1406; Sigma-Aldrich), anti-GAD67 (MAB5406; Chemicon), anti-glutamate (AB133; Chemicon), anti-synaptophysin (MAB368; Chemicon), anti-PSD95 (04-1066; Millipore), and anti-PH3 (06-570; Upstate). Secondary antibodies were used as follows: Alexa Fluor 488 goat anti-mouse (A11001; Invitrogen) and Alexa Fluor 546 goat anti-rabbit (A11010; Invitrogen). DAPI (4′, 6-diamidino-2-phenylindole, 1 mg/mL) was used for nuclei staining. Images were acquired using a high content automated microscope (Operetta; Perkin Elmer).

Positive cells for GAD67 and phosphorylated histone 3 (PH3) staining were quantified in the entire section of cerebral organoids. To quantify glutamate intensity, mean gray value (fluorescence intensity) was measured in three points of each cerebral organoid border, delimited by a rectangular selection. The fluorescence intensity in the cerebral organoids’ edge was normalized for the tissue background and was given as fold increase in basal condition. The numbers of 30 and 45-days independent samples for quantifications were as follows: PH3 positive cells, n = 3 and n = 5; MAP2 area, n = 4 and n = 6; GAD67 positive cells, n = 2 and n = 4; glutamate fluorescence intensity, n = 6 and n = 7, respectively.

Synchrotron radiation X-ray fluorescence (SR-XRF) spectroscopy analysis

For XRF analysis, cerebral organoids were quickly rinsed in PBS, embedded in OCT and frozen. Organoids were cut into 30 µm thick sections and placed on ultralene film (transparent to X-ray) and air-dried. Four tissue sections of independent 30-days old cerebral organoids and five sections of independent 45-days old cerebral organoids were analyzed.

The SR-XRF analyses were performed at the D09B X-ray fluorescence beamline at the Brazilian Synchrotron Light Source (Pérez et al., 1999) (Campinas, Brazil) using standard temperature and pressure conditions. Samples were excited by a white beam with energy ranging from 5 keV to 17 keV. An optical system based on a pair of bent mirrors in a Kirkpatrick-Baez arrangement was used to focus the X-ray beam down to 20 µm spatial resolution. Each spot was irradiated for one second. A silicon drift detector (KETEK GmbH) with 140 eV (FWHM) at 5.9 keV placed at 90° from the incident beam was used to collect X-ray fluorescent and scattered radiation coming from samples.

Concentration values in weight fraction units for the elements detected in each pixel of the cerebral organoid slice were determined by the PyMCA software developed by the Software Group of the European Synchrotron Radiation Facility (Solé et al., 2007). After calibrating the spectrometer with a set of pure thin films from Micromatter™ standards (http://www.micromatter.com), the fundamental parameter method was used (He & Vanespen, 1991). The method predicts the sample matrix effect assuming that the measured primary spectral distribution and excitation-detection geometry are known. The values found in weight fraction units were converted in ppm by multiplying them by a factor 106. The mean concentration value was calculated with the values detected in each pixel analyzed per organoid slice.

Image analyses of trace elements distribution

Elemental concentration information was collected from each pixel to generate corresponding XRF maps. The color gradient heat maps were constructed by plotting the fluorescence intensity at each point of the scan using the PyMca software.

Images of colored gradient heat maps were converted to grayscale images with Adobe Photoshop CS5 by manually adjusting the hue to match the colors’ intensity in RGB images. Then, ImageJ software was used to obtain the intensity profile by drawing a line in sagittal and coronal planes of the organoid images. The generated plot profiles were analyzed in Igor Pro 6 software (Wavemetrics, Lake Oswego, OR, USA) to create curve fittings with polynomial regression of three terms. The resulting curves were classified as concave or convex. If the fitting result gave rise to a line with less than 10 points of inclination, it was considered as a straight line. We assumed concave as peripheral, convex as central and straight as diffused distribution of the elements within cerebral organoids. Four tissue sections of independent 30-days old cerebral organoids and five sections of independent 45-days old cerebral organoids were analyzed.

Elemental correlation analysis

In order to unveil elements correlation we have used PyMca software to build dot plots combining concentration data for all possible pairs of elements in each scanned pixel. Pearson’s correlation coefficient was applied to each element pair, distinguishing correlated and uncorrelated elements. In addition to Pearson’s correlation coefficient, we also fitted a line to each dot plot, generating a R-squared value for each pair of elements. This number was also used as a correlation indicator. A high R-squared above 70% was interpreted as indicator of potentially meaningful associations in paired elements. Finally, color-gradient merged maps were also built using PyMca software, whereas color colocalization indicated elements higher correlation, distinguishable colors were an indicative of unrelated or exclusive elements. Four tissue sections of independent 30-days old cerebral organoids and five sections of independent 45-days old cerebral organoids were analyzed.

Statistical analyses

Quantitative data were reported as mean ± S.D. Analyses of statistical significance were obtained using GraphPad Prism 4 software (GraphPad Software, La Jolla California, USA). Comparisons among organoids at 30 and 45 days of differentiation were conducted with unpaired t-test. Comparisons among organoids at seven, 15, 30, and 45 days of differentiation were analyzed using one-way ANOVA followed by Tukey’s post hoc test.

Results

The growth of human cerebral organoids

Exogenous organogenesis using pluripotent stem cells has emerged as a breakthrough technology to study aspects of human brain development in a dynamic and living state (Lancaster et al., 2013). In our preparations, cerebral organoids’ differentiation from human embryonic stem cells occurred in spinner flasks as represented in Fig. 1A and, by the end of 45 days of differentiation, they presented spheroid morphology (Fig. 1B). In detail, 30-days old organoids demonstrated different hues and internal folding, suggestive of distinct cell layers (Figs. 1C and 1D). A period of accelerated growth was observed between the 7th and the 15th day, followed by a period of stationary growth upon retinoic acid addition (15th to 30th day). Then, cerebral organoids continued to expand in size from day 30 to day 45, when they reached 1,240 ± 365 µm in diameter (Fig. 1E). Cerebral organoids derived from iPS cells also grew in a similar fashion (Figs.  S2A–S2C).

Figure 1 Cerebral organoids derived from human embryonic stem cells.

(A) Sequential steps involved in the generation of cerebral organoids from human pluripotent stem cells. (B) Spheroid 45-days old cerebral organoids presenting smooth texture and homogeneous coloring. (C and D) Microphotography of a 30-days old organoid in detail showing distinct hues according to different cell layers. (E) Organoid diameter quantification; organoids were measured at different stages of the differentiation process. Organoids’ diameter doubled between days 7 and 15 in culture and quintupled after 45 days. The graph represents mean ± S.D. n = 107 for 7-days old organoids, n = 90 for 15-days old organoids, n = 56 for 30-days old organoids, n = 18 for 45-days old organoids. ∗∗∗p < 0.0001 for 7-days old versus 15, 30 and 45-days old organoids. Cerebral organoids were obtained from two independent assays. Scale bars: B = 1 mm, C = 125 µm and D = 25 µm.

As one of our major goals was to generate a cortical anlage from human embryonic stem cells in vitro, we then examined the cytoarchitecture of the generated cerebral organoids. We focused on two separate time points, 30 and 45 days of differentiation, which corresponded to 15 and 30 days, respectively, of exposure to retinoic acid, a crucial morphogen for neuronal induction derived from vitamin A. Upon 30 days of differentiation, prominent circular structures reminiscent of early ventricles were observed within the organoids (Figs. 2A and 2B). Interestingly, following organoids’ maturation, the large epithelium-lined cavities were replaced by narrower ones (Figs. 2C and 2D) of reduced luminal area (Fig. 2E). Also, the number of putative ventricles or epithelium-lined cavities per cerebral organoid decreased from 30 to 45 days of differentiation (Fig. 2F).

Figure 2 Cerebral organoids present epithelium-lined cavities morphologically similar to ventricles.

(A) Within 30 days of differentiation, cerebral organoids formed large cavities morphologically similar to ventricles, lined with stratified epithelium (B). (C) After 45 days of differentiation, the lumens of these cavities seemed flattened and (D) the surrounding epithelium length seemed reduced. (A, B, C, and D) H&E staining. (E) Measurements of the putative ventricles showed a reduction in average luminal area. (F) Concomitantly, the number of epithelium-lined cavities per cerebral organoid tissue section was reduced from 30 to 45 days of differentiation. The graphics represent mean ± S.D. Luminal area measurements: n = 17 for 30-days old organoids and n = 09 for 45-days old organoids. Number of putative ventricles per organoid: n = 6 for 30-days old organoids and n = 11 for 45-days old organoids, ∗p < 0.05. Cerebral organoids were obtained from two independent assays. Scale bars: A and C = 100 µm, B and D = 50 µm.

Besides substantial differences in size and architecture, we further investigated changes on the expression of neural markers via immunofluorescence. Thirty days into neural differentiation, cells expressing the intermediate filament nestin, characteristic of CNS progenitors, were ubiquitously found in the developing organoids (Fig. 3A). Similar to the cortical development in vivo, areas immediately adjacent to the ventricle-like cavities had greater cell density and exhibited a radial and outward polarization suggestive of zones of cellular division and migration (Fig. 3B). Indeed, the presence of apical progenitors was characterized by mitotic cells lining the luminal surface, positively stained for PH3 (Fig. 3B), and by the expression of the transcription factor PAX6 in the putative ventricular zone (Fig. 3C). Furthermore, intermediate progenitors expressing the T-box homeobox protein TBR2 were found midway to the putative subventricular zone (SVZ), positioned radially to the luminal surface and adjacent to tangentially migrating neurons (MAP2 positive cells) (Fig. 3D) in a cellular architecture similar to the developing cortical plate in vivo (Lui, Hansen & Kriegstein, 2011). Likewise, in iPS cells derived cerebral organoids, differentiated neurons were observed positioned in the outer rim of the cellular layer around the putative ventricles, whereas neural progenitors occupied the innermost portion (Figs. S2D–S2F).

Figure 3 Cerebral organoid cytoarchiteture after 30 and 45 days in culture.

In 30-days old organoids, (A) positive cells for the intermediate filament nestin were observed throughout cerebral organoids’ extension. (B) The luminal surface of the epithelium-lined cavities was populated by mitotic cells (PH3), (C) identified as apical progenitors (PAX6) in the ventricular zone. (D) Intermediate progenitors expressing TBR2 were also present in the ventricular zone and composed an adjacent layer, the subventricular zone. (C and D) Tangential migratory neurons (β-tubulin III and MAP2) established the pre-plate outside germinal zones. In 45-days old organoids, (E) the flattened ventricles still presented proliferative cells (PH3) and (F) neural progenitors (TBR2) in the radially organized cell layer. (G) MAP2 positive cells were found widespread, except in germinal zones, while GAD67 (GABA synthesis enzyme) positive cells (H) or the neurotransmitter glutamate (I) became evident. In this time point, synaptic markers such as synaptophysin and PSD95 (J) were also observed. Finally, comparing 30 to 45-days old cerebral organoids, there was a reduction in the number of PH3 positive cells (K), while the neuronal population was expanded (L) as well as neurons producing GABA (M) or cells producing glutamate (N). Graphics are represented as mean ± S.D. For 30-days old and 45-days old organoids, respectively: PH3 positive cells, n = 3 and n = 5; MAP2 positive area, n = 4 and n = 6; GAD67 positive cells, n = 2 and n = 4; glutamate fluorescence intensity, n = 6 and n = 7, ∗p < 0.05. Scale bars: A, B, F, G = 100 µm; C, D, E, H, J = 50 µm.

In 45-days old organoids, flattened cavities still presented proliferating cells (PH3 positive cells) and intermediate progenitors (TBR2) in its vicinity, showing their commitment to VZ/SVZ identity (Figs. 3E–3F). Mature neurons identified by MAP2 staining demonstrated the production of a neuronal primordial plexiform layer organized tangentially to germinal zones (Fig. 3G). Migratory GABAergic neurons were detected by glutamic acid decarboxylase 67 (GAD67) staining in different regions of the organoids, primarily near the ventricles (Fig. 3H). Concurrently, glutamate was detected in the organoids’ edge (Fig. 3I). Finally, synaptogenesis was determined by the presence of synaptophysin, a component of presynaptic vesicles, and also by the detection of the postsynaptic density protein 95 (PSD95) (Fig. 3J).

As a result of the transitioning from a main self-renewal stage to a neuronal differentiation phase, a five times decrease in mitotic activity was observed when comparing the organoids at 30 and 45 days of differentiation (Fig. 3K). Accordingly, the amount of mature neurons tripled, as evaluated by MAP2 staining (Fig. 3L), and the number of neurons expressing GAD67 increased 4 times (Fig. 3M), when comparing organoids from the 30th to the 45th day of differentiation. In addition, peripheral glutamate staining increased 2.5 times (Fig. 3N). In line with these data, we considered these two time points (30 and 45 days) to depict two distinct developmental phases: one of pronounced cell division dedicated to tissue expansion and another of early neuronal network formation in cerebral organogenesis. Then, we asked whether these two time points might show different element distribution as they represent two demarcated developmental stages.

SR-XRF microprobe analysis

SR-XRF microprobe was used to scan elements in cerebral organoids. After X-rays excited the sample, each atom emitted a unique, identifiable and quantifiable photon signature. XRF analysis revealed that the major chemical elements in cerebral organoids were phosphorus (P), sulfur (S), potassium (K), calcium (Ca), iron (Fe) and zinc (Zn). The elements manganese (Mn), nickel (Ni) and copper (Cu) were also detected in our preparations, but were not considered further in this study due to their extremely low levels and subsequent classification as ultratrace elements. Although chlorine (Cl) was detected, it was disregarded for further analysis as it was considered a potential laboratorial artifact. While specific roles and importance of the detected elements for brain development or function may be under characterization, some of the known functions are summarized in Table 1.

Table 1 Trace elements detected in cerebral organoids and its corresponding functions in the brain development.

Element	Cellular function	Role in brain development	References	
P	•  Nucleoproteins
•  Phospholipids
•  ATP	•  Neurogenesis (cellular membrane biogenesis)	Silvestre, Maccioni & Caputto (2009)	
S	•  Protein synthesis
•  Cysteine
•  Disulfide bonds	•  Neurogenesis
•  Neuronal migration (proteoglycans and glycosaminoglycans)
•  Axon guidance	Inatani et al. (2003), Girós et al. (2007) and Maeda (2015)	
K	•  Protein synthesis
•  RNA synthesis
•  Cell division
•  Transmembrane transport	•  Resting potential
•  Neurogenesis
•  Neural progenitors’ survival	Lubin & Ennis (1964) and Schaarschmidt et al. (2009)	
Ca	•  Cell signaling
•  Enzymatic cofactor
•  Regulation of gene expression
•  Cell motility	•  Neurogenesis
•  Neuronal differentiation
•  Release of synaptic vesicles	Weissman et al. (2004), Shin et al. (2010), Resende et al. (2010) and Atlas (2013)	
Fe	•  DNA synthesis
•  Lipid synthesis
•  Energetic metabolism
•  Metalloproteins	•  Neurogenesis
•  Neuronal differentiation
•  Monoamine metabolism
•  Modulation of dopaminergic receptors
•  Synaptic maturity
•  Myelin synthesis	Erikson et al. (2001), VanLandingham & Levenson (2003), Lozoff & Georgieff (2006) and Tran et al. (2008)	
Zn	•  Protein synthesis
•  DNA synthesis
•  Zinc finger transcription factor
•  Metalloproteins	•  Neurogenesis
•  Neuronal differentiation
•  Synaptic modulation
•  Dendritic arborization
•  Myelin synthesis	Liu et al. (1992), Gao et al. (2009), Levenson & Morris (2011), Morris & Levenson (2013) and Marger, Schubert & Bertrand (2014)	

Trace elements distribution in human cerebral organoids

In order to search for elemental distribution patterns in cerebral organoids, the spectral profile corresponding to each measurable element was collected and assembled into color-gradient heat maps, as represented in Fig. 4. Phosphorus could be detected all over organoids’ extension, despite being concentrated within the external border in both 30 and 45-days old organoids. Besides being observed diffusely distributed as well, K levels were higher in the organoids’ edge in 30 days and then became evenly distributed in 45 days. Sulfur, Ca and Fe displayed a more homogeneous distribution pattern in both proliferative and neuronal differentiation phases. Meanwhile, Zn was mostly diffused in 30-days old organoids and then became peripheral in 45-days old organoids. Elementary pattern distributions revealed a tendency for diffuseness for most elements, with the exceptions of P and Zn. Rather than spreading internally, these two elements continued to be located in higher concentrations in the edge of cerebral organoids.

Figure 4 Elemental distribution in cerebral organoids.

Representative color gradient heat maps of 30 and 45-days old cerebral organoids. Elemental maps revealed diffusion patterns for P, S, K, Ca, Fe and Zn. Scales correspond to weight fraction units (ppm or µg/g = weight fraction∗106).

Elemental concentration in cerebral organoids

To compare the contribution of elements in, here named, proliferative (30 days) and neuronal maturation (45 days) stages, we estimated concentrations of trace elements in parts per million (ppm), which are shown in Fig. 5. Phosphorus, S and K were the most abundant elements in both 30-days (16,142 ± 1,219 ppm; 4,955 ± 350 ppm and 6,120 ± 1,745 ppm, respectively) and 45-days old organoids (10,286 ± 840 ppm; 4,462 ± 249 ppm and 3,951 ± 422 ppm, respectively). Calcium, Fe and Zn were found in relatively low levels: 192 ± 157 ppm, 84 ± 61 ppm and 129 ± 6 ppm in 30-days old organoids, and 286 ± 43 ppm, 87 ± 37 ppm and 101 ± 2 ppm in 45-days old organoids, respectively. Interestingly, some trace element levels decreased from 30 to 45 days of differentiation, such as P, K and Zn, whereas Fe tended to increase, even though, this latter result did not reach statistical significance.

Figure 5 Elemental quantification in cerebral organoids.

Cerebral organoid elemental composition was also assessed by SR-XRF. Scatter plots represent values found for each sample. (A) P, ∗∗∗p = 0.0001; (B) S, p = 0.066; (C) K, ∗p = 0.05; (D) Ca, p = 0.29; (E) Fe, p = 0.92; (F) Zn, ∗∗∗p < 0.0001. PPM, parts per million.

To determine to which degree the presence of trace elements in cerebral organoids could be due to passive diffusion from medium or to active cellular metabolism, we compared elements’ availability in cell culture media and in Matrigel with the amounts found within the organoids (Tables S1 and S2). These quantifications revealed that both cell culture media and Matrigel contributed very poorly to total organoid elemental quantification, being responsible for approximately 0.002% and 0.0000000004% of the amount of trace elements detected in the cerebral organoids, respectively. This substantial disparity strongly suggests that the increased levels of chemical elements observed in cerebral organoids were due to its biological constitution and did not reflect a simple diffusion mechanism.

Inter-elemental relationship

From an initial inspection of the spatial distribution of each element, it was not possible to assess, with any confidence, to which extent they may interact functionally or chemically, as would occur when they become part of biomolecules. We then analyzed elements in pairs to assess coincidental areas, classifying elements as correlated, unrelated or exclusive. Table 2 comprises correlation analyses of all possible elemental pairs. We have analyzed elements by Pearson’s coefficient correlation and by R-squared values obtained from lines fitted on raw quantification data. Correlation coefficients closer to 1 indicate higher elemental correlation, while lower correlation coefficients indicate lower elemental correlation. P/S, P/K, P/Zn and S/K were highly correlated elements, as they presented Pearson’s correlation coefficients above 0.7 in average. On the other hand, P/Ca, P/Fe, S/Ca, S/Fe, K/Ca, K/Fe, K/Zn, Ca/Zn, Fe/Zn were less correlated pairs of elements, presenting Pearson’s correlation coefficients lower than 0.7. Interestingly, the following pairs of elements: P/Ca, S/Ca and K/Ca presented great changes in their correlation coefficients from 30 to 45 days of development (Table 2, 45/30 days ratio), indicating that their behavior might change in a time-dependent manner. Figure 6 comprehends representative color-gradient maps for P/S, P/Zn and K/Ca, highlighting their behavior. Although P/S present a high correlation coefficient, they seem to partly lose colocalization at 45 days (Figs. 6A and 6D), which could be seen by elemental distribution within the map. P/Zn (Figs. 6B and 6E ) presented a similar behavior. Finally, K/Ca (Figs. 6C and 6F) presented a very interesting pattern: Ca was particularly localized at 30 days, and evolved to a scattered distribution at 45 days of differentiation.

Table 2 Elemental correlation analysis for pairs of elements.

Pearson’s product-moment correlation coefficient (Pearson’s r) and R-squared from lines of best fit were calculated for each elemental pair in order to reveal correlated and uncorrelated pairs. A 45 days/30 days old organoids ratio was calculated in order to highlight time-dependent changes in elements’ correlational distribution. Data are represented as mean ± S.D. For 30-days old cerebral organoids, n = 4 and for 45-days old cerebral organoids, n = 5. For all correlations of element pairs, p < 0.05; except for one 30-days old cerebral organoid in the analysis of K/Fe pair and one 45-days old cerebral organoid in the analyses of K/Fe and P/Fe pairs.

Compared elements (pairs)	Pearson’s r	R-squared	
	30 days (mean ± S.D.)	45 days (mean ± S.D.)	45/30 days (ratio)	30 days (mean ± S.D.)	45 days (mean ± S.D.)	45/30 days (ratio)	
P/S	0.9607 ± 0.0078	0.9266 ± 0.0097	0.96	0.9231 ± 0.0149	0.8587 ± 0.0180	0.93	
P/K	0.8345 ± 0.0826	0.8922 ± 0.0112	1.07	0.7016 ± 0.1363	0.7962 ± 0.0199	1.13	
P/Ca	0.1927 ± 0.1791	0.5509 ± 0.2528	2.86	0.0612 ± 0.0810	0.3546 ± 0.2263	5.79	
P/Fe	0.4294 ± 0.4468	0.3942 ± 0.3071	0.92	0.3341 ± 0.3689	0.2309 ± 0.2449	0.69	
P/Zn	0.8365 ± 0.0385	0.7331 ± 0.0467	0.87	0.7008 ± 0.0644	0.5391 ± 0.0664	0.77	
S/K	0.8490 ± 0.0794	0.9358 ± 0.0157	1.10	0.7256 ± 0.1315	0.8759 ± 0.0292	1.21	
S/Ca	0.1851 ± 0.1592	0.5951 ± 0.2744	3.21	0.0532 ± 0.0665	0.4144 ± 0.2662	7.79	
S/Fe	0.4386 ± 0.4522	0.5000 ± 0.2286	1.14	0.3457 ± 0.3811	0.2892 ± 0.2317	0.83	
K/Ca	0.1484 ± 0.1225	0.5979 ± 0.2638	4.03	0.0333 ± 0.0404	0.4132 ± 0.2649	12.41	
K/Fe	0.2944 ± 0.3534	0.4018 ± 0.2987	1.36	0.1551 ± 0.2437	0.2283 ± 0.2294	1.47	
K/Zn	0.7137 ± 0.0674	0.6493 ± 0.0777	0.91	0.5128 ± 0.0968	0.4265 ± 0.0989	0.83	
Ca/Zn	0.3459 ± 0.1044	0.4256 ± 0.1003	1.23	0.1278 ± 0.0744	0.1891 ± 0.0784	1.48	
Fe/Zn	0.5114 ± 0.3433	0.4445 ± 0.1812	0.87	0.3500 ± 0.3418	0.2239 ± 0.1791	0.64	

Figure 6 Concentration gradient heat maps overlay for elemental colocalization in cerebral organoids at 30 and 45-days of development.

Representative merged color-gradient maps of organoids’ elements pairs: P/S at 30 (A) and 45 (D) days of differentiation; P/Zn at 30 (B) and 45 (E) days of differentiation; K/Ca at 30 (C) and 45 (F) days of differentiation. Elements are represented in: P (red), S (green), K (blue), Ca (red) and Zn (blue).

Discussion

Implications of trace elements detected in cerebral organoids to brain organogenesis

While it is generally accepted that micronutrients are vital to brain morphogenesis, neurochemistry and neurophysiology, well-controlled studies for specific micronutrients are still needed. Furthermore, since most studies on this subject have been done on adult-state or post-mortem tissue samples, little is known about the elemental composition of the developing human CNS. Though it is not an exact replica of a human embryonic brain, exogenously developed cerebral organoids undergo many developmental stages and events that parallel the human condition. Perhaps more importantly, the circuits and structures being constructed contain a significant portion of the human genetic blueprinting and specific neurons that make up the early neuronal networks that give rise to the brain (Lancaster et al., 2013). Therefore, the model used in this study, amongst others of human organogenesis, could be argued to be the closest and most complete study system to date for understanding human neural development and its pathological manifestations.

The methodology for the development of cerebral organoid tissues described in here subtly deviates from the original method reported by Lancaster et al. (2013). Specifically, the aggregation of dissociated pluripotent stem cells into EB was conducted in spinner flasks instead of using individualized non-adherent 96-well plates. The EB obtained here with iPS cells presented similar sizes when compared to Lancaster’s descriptions for those derived from the H9 (WA09) hESC line (Lancaster & Knoblich, 2014). However, EB obtained from the BR1 hESC line were relatively smaller. One possibility is that these results may reflect inter-lineage variability in stem cells aggregation potential, as already described by others (Cahan & Daley, 2013). Importantly, regardless of such difference in EB size, by the end of 30 and 45 days of differentiation, the cytoarchitecture of cerebral organoids’ was similar to the original report (Lancaster et al., 2013).

SR-XRF is a suitable technique to scan and image trace elements in brain tissue and has been applied in many models of neurological diseases. The primary advantages in using SR-XRF lie in its highly sensitive detection, topographic high-resolution chemical imaging and recognition of metal compounds irrespectively of its oxidation state. Given that metal metabolism is disrupted in prevalent neurodegenerative disorders, such as Alzheimer’s, Huntington’s, and Parkinson’s diseases (Popescu et al., 2009; Wang et al., 2012; Muller & Leavitt, 2014), SR-XRF can also be applied for a comprehensive view of metal homeostasis in brain development and aging. Contrary to the research aims carried out by other groups, our study intended to employ SR-XRF on human cerebral organoids to provide us with the first glimpse into which elements may play active roles in early brain development. More specifically, we reported here that P, S, K, Ca, Fe and Zn take part in neural composition during cerebral organoid formation.

Phosphorus was the most heavily represented element. When comparing proliferative (30 days) and neuronal differentiation (45 days) stages, we were able to find a decrease in P levels. Since P is a structural component of major biomolecules, such as nucleotides and phospholipids, this reduction might be explained by a switch from a phase of intense synthesis, including DNA and phospholipid production, to a more migratory and differentiation phase. With respect to tissue growth, P levels in cerebral organoids were in the same concentration range (mg/g) of that reported for adult brains (Rajan et al., 1997), albeit having been noted here in a higher quantity (Fig. 5A). In this regard, the levels found in our study imply a greater participation of P during the development of neural tissue (Rajan et al., 1997). Silvestre and colleagues (2009) also showed that total lipid P content is higher in the embryonic brain than in adult brain cells. The developing brain contains superior concentrations of phosphate groups belonging to lecithins, cephalins, and sphingomyelins, this could explain the different values found in our study. This dissimilarity underscores the need for pursuing elemental changes at either end of the aging spectrum.

Cerebral organoids were also shown to contain significant levels of K and S. Potassium is essential to transmembrane transport, regulation of cellular volume, and membrane resting potential. During brain ontogenesis, K regulates proliferation of neural progenitors (Achilles et al., 2007; Yasuda, Bartlett & Adams, 2008; Schaarschmidt et al., 2009) and is maintained in heightened levels in rapidly dividing cells (Cameron, Pool & Smith, 1979; Wallberg et al., 2000). As K currents also support the migration of early neurons over long distances due to cellular volume changes (Hendriks, Morest & Kaczmarek, 1999), we can speculate from our data that this element could be important to maintain mitotic activity in 30-days old organoids and to promote cell migration in 45-days old organoids, amongst other roles.

It is known that S integrates virtually all proteins through methionine and cysteine amino acids and disulfide bounds. In the embryonic cerebrum, chondroitin and heparan sulfate proteoglycans anchor attractive and/or repulsive cues such as growth factors, chemokines, axon guidance molecules, and cell adhesion molecules important for neuronal migration in strategically routes such as the striatum, marginal zone, subplate, and subventricular zone in the neocortex (Maeda, 2015). As an example, brains devoid of perlecan, a heparan sulfate proteoglycan component of the CNS extracellular matrix, have impaired cortical development and are microcephalic (Girós et al., 2007). Beyond this, S is also incorporated in proteoglycans expressed in the basal lamina of the neuroepithelium that regulates neurogenesis in the developing telencephalon (Girós et al., 2007). In cerebral organoids, we found high levels of S, both in 30 and 45 days of differentiation, pointing to its central role in organoids’ patterning. Therefore, we postulate that S may play a fundamental role in brain construction and in organoids’ scaffolding and patterning.

The metallic elements Ca, Fe, and Zn, which are essential for brain morphogenesis, were found in the µg/g range in cerebral organoids. Calcium participates in a variety of cellular functions in different areas extending from proliferative zones to post mitotic intermediate zone and marginal zone. In neural progenitor cells, Ca waves, sustained by IP3-signaling, control cell proliferation (Weissman et al., 2004; Resende et al., 2010), while Ca influx via ion channels influences cell differentiation (Shin et al., 2010). Importantly, during early neurogenesis, the expression of transcription factors regulating neuronal survival and differentiation can be controlled by Ca (Leclerc et al., 2012). Noteworthy, Ca ppm values detected in the present work are consistent with those found by Riederer et al. (1989) and Rajan et al. (1997) in human cerebrum cortex (Riederer et al., 1989; Rajan et al., 1997) (Fig. 5D and Table S3). Altogether, these findings emphasize that Ca may pave the way to brain organogenesis.

Zinc is an essential element to protein synthesis, enzymatic catalysis and serves as a structural component to zinc finger transcription factors. Zinc-dependent enzymes include metalloproteinases and many intermediary metabolism dehydrogenases important for CNS function (Tapiero & Tew, 2003). In addition, chelatable Zn present in synaptic vesicles can be released in the synaptic cleft to modulate many synaptic activities (Marger, Schubert & Bertrand, 2014). In comparison to previous studies in adult brain, we found higher levels of Zn in cerebral organoids (Katoh, Sato & Yamamoto, 2002; Rahil-Khazen et al., 2002), although compatible in scale of dosage (Fig. 5F). While greater concentrations of Zn in adult brain are toxic to neural cells and also cause axonal degeneration in mice (Chuah, Tennent & Jacobs, 1995), Zn modulates stem cell proliferation and neuronal differentiation during neurogenesis (Gao et al., 2009; Levenson & Morris, 2011; Morris & Levenson, 2013). In accordance, previous data from our group have demonstrated increased Zn during neural differentiation of human pluripotent stem cells (Cardoso et al., 2011). This may highlight the importance of Zn for neural stem cells commitment and justify why the levels found in cerebral organoids are subtly higher than those described in adult brain (Fig. 5F and Table S3) (Dexter et al., 1991; Rajan et al., 1997; Rahil-Khazen et al., 2002).

It is currently known that Fe is accumulated during brain prenatal development, with the highest levels observed immediately after birth. In the developing brain, Fe is required in rapidly developing regions such as the cerebral cortex (Siddappa et al., 2003) in heme-containing cytochromes that regulate neuronal and glial energetic status (Evans & Mackler, 1985). Along the same line, Fe is required to the initial expansion of neural tissue and by neurogenesis, which may account for its levels in the present study. It is noteworthy that the Fe values found for cerebral organoids parallel those measured in the adult brain (Sofic et al., 1988; Dexter et al., 1991; Rajan et al., 1997; Rahil-Khazen et al., 2002). Although higher Fe concentrations are found in some specific cerebral regions such as substantia nigra, putamen and globus pallidus (Dexter et al., 1991), they were not represented in this study. As examples of Fe roles in the CNS, Fe-containing enzymes are essential for tyrosine and tryptophan hydroxylase activities and for monoamine catabolism in synapses. Besides, Fe is also required for ribonucleotide reductase in order to regulate cell division and for normal myelination throughout CNS development (Lozoff & Georgieff, 2006).

Overall, cerebral organoids derived from pluripotent stem cells presented an elemental composition mostly compatible with previous reports (Fig. 5 and Table S3). However, some discrepancies can be found, such as Fe levels. We have found approximately 90 ppm, while values reported for fresh harvested autopsies are still diverging, including 225 ppm in cerebral cortex in the motor field (Katoh, Sato & Yamamoto, 2002), 101 ppm in the precentral gyrus (Popescu & Nichol, 2011), and 50.2 ppm in the brain front lobe (Rahil-Khazen et al., 2002). Considering that these studies were performed on adult tissue samples, while cerebral organoids generated in vitro are an embryonic model, these differences can be attributed to sample origin. Another possibility is that in vivo models surpass elemental concentrations observed in in vitro models, as already reported for murine cultured neurons and astrocytes (Hare et al., 2013). Nevertheless, we believe that proper organoid quantifications are relevant, especially when comparing healthy vs. pathological conditions, such as in our work on iPS cells derived from schizophrenic patient (Paulsen et al., 2014).

Analysis of different organs also revealed similarities and differences from neural tissues and our cerebral organoids. Reports on Fe content in breast (∼5 ppm) (Geraki, Farquharson & Bradley, 2002), heart (257 ppm), kidney (430 ppm), liver (837 ppm), lungs (987 ppm) (Katoh, Sato & Yamamoto, 2002), among other organs, reveal actual variation intervals (5 to 987 ppm). With 3D organoid technology boosting in vitro human organs modeling, we believe that elemental characterization will be a relevant tool for assessing general and discrete processes within these models, in addition to revealing chemical patterns and enabling in vivo vs. in vitro and/or healthy vs. pathologic state comparisons.

Trace elements distribution in human cerebral organoids

Our data indicate that trace elements change their distribution inside organoids along differentiation. It is known that some molecules can pass through cell membrane by simple diffusion or with the aid of specific cellular transporters or channels. Potassium was reported as firstly concentrated at the organoids’ border and then became evenly distributed inside. Since K is positively charged and therefore not permeant to cell membranes, one possibility is that NKCC1, an inwardly directed cotransporter, begins to be expressed and might contribute to the simultaneous K and Cl influx. NKCC1 expression is upregulated in early developmental stages and downregulated in later brain development in vivo (Kaila et al., 2014).

Even though the distribution profile of a given element may be associated with changes in the expression of specific receptors and/or channels in the organoid, one can also suggest a particular role for such element in a specific cellular niche. For example, Zn displayed a more homogeneous distribution within organoids, but became relatively more peripheral at 45 days. Of note, glutamate was almost undetectable in organoids at 30 days of differentiation, but at 45 days, it occupied the outer portion of the organoids. It is known that Zn is present at high levels in glutamatergic synaptic vesicles of forebrain neurons and that its influx can be mediated by AMPA channels (Martínez-Galán, Díaz & Juiz, 2003; Takeda et al., 2009). Beyond this, Zn-dependent metalloproteinases facilitate neural migration and regulate neurite outgrowth in the preplate and cortical plate (Sanz, Ferraro & Fournier, 2015; Sîrbulescu, Ilieş & Zupanc, 2015). Hence, one possibility is that Zn intensity matches cellular distribution of glutamatergic cells and neuronal synaptogenesis. In this regard, further studies should help identify changes in the expression of specific channels and receptors that may take part in microelemental homeostasis of cerebral organoids.

There is another very important feature that has to be taken into account when discussing element distribution in cerebral organoids: cell density. Both organoid groups, 30 and 45 days of differentiation, presented increased cell density in organoids’ edge. In 30-days old organoids, edges had 3.9 times more cells than the center, while in 45-days old organoids this difference decreased to 1.3 times (Data S8). This behavior may explain the fact that K was mainly present near the edge in 30-days old organoids and became more diffuse in 45-days old organoids. Nevertheless, cell density did not influence S, Ca, and Fe distribution, as these elements do not followed K pattern. Also, Zn was only concentrated near the edge at 45 days of development.

Inter-elemental correlations in human cerebral organoids

The atomic elements can exist freely intra and extracellularly in the form of ions or ionic groups, and in association with biomolecules, for example. These associations define or, are defined by cellular processes, in which two or more elements behave in similar ways. Associations of this kind can be investigated by elemental correlation studies. Our analysis revealed interesting patterns, as P, S, K and Zn seem to be highly correlated, whereas Ca and Fe present very low correlation levels. In addition, we also found that P, S, Zn, K and Ca change their correlation pattern from 30 to 45 days of differentiation. These phenomena are interesting as they may portrait the change from a proliferation to a neuronal maturation stage during organoid development.

Zinc is not only a structural element, but also acts as a regulator of cell proliferation. Based on studies with Zn chelators in mammalian cells, it was found that Zn deficiency results in reduced expression of thymidine kinase (Chesters, Petrie & Travis, 1990) and reduction in thymidine incorporation (Chesters, Petrie & Vint, 1989). Strictly, Zn integrates DNA polymerase (Springgate et al., 1973), RNA polymerase (Wu et al., 1992), and ribosomal proteins (Härd et al., 2000). Whilst purely speculative, it is possible that P, as being part of nucleic acids and present in proteins, shares the same location with Zn, as both are involved in cell proliferation and protein synthesis. Another interesting example is K, which seems to be directly involved in the control of protein synthesis (Lubin & Ennis, 1964) and, in conjunction with P, is enriched in ribosomes. Specifically in neurons, K tends to correlate with P in the adult rat brain (Cameron, Sheridan & Smith, 1978). In a previous work from our group, we have shown that neurospheres generated from iPS cells derived from schizophrenia patient biopsies present higher levels of Zn and K, while producing high levels of reactive oxygen species (ROS) (Paulsen et al., 2014). Despite the fact that these studies did not involve direct correlation between metals and biomolecules, we were able to link this correlation to a disease model involving Zn defective transport leading to K imbalance. In sum, we conclude that element correlation is a fundamental part of SR-XRF analysis, especially taking into consideration the organoid model, since its cellular organization is much more complex than EB and neurospheres.

Functional significance

Mounting evidence indicates that maternal malnutrition may be causative in many neurocognitive deficits and neurological diseases in offspring (Felt & Lozoff, 1996; De Souza, Fernandes & Do Carmo, 2011). Given that maternal diet is the main source of dietary elements available to a developing fetus; our data highlight the importance of matching the essential elements P, S, K, Ca, Fe and Zn to developmental needs of the brain. For instance, deficiencies in Zn nutrition during prenatal development are associated with offspring learning and memory paucity (Liu et al., 1992; Tahmasebi Boroujeni et al., 2009; Yu et al., 2013), in part caused by decreased expression of brain-derived neurotrophic factor (BDNF), altered myelin composition (Liu et al., 1992) and declined long-term potentiation (Yu, Ren & Yu, 2013). Gestational Fe anemia, the most common nutritional need, can impact learning, memory, and motor abilities in progeny. These poor executive performances can be explained at the cellular level by decreased synaptic maturity, dopamine metabolism and myelin composition (Lozoff & Georgieff, 2006). In a prospective study, it was found that maternal Fe deficiency may be a risk for schizophrenia in offspring (Insel et al., 2008). This means that the diet influences pregnancy and, to some extent, can have long-term consequences in fetal brain structuring.

Conclusions

Here, we have shown the potential of cerebral organoids in conjunction to XRF analysis to explore minerals homeostasis during brain development. Mapping each micronutrient could be useful for indicating the expression of specific receptors and/or channels, as well as for locating elements that take part in neural composition during cerebral organogenesis, such as P, S, K, Ca, Fe and Zn in a particular cellular niche. Also, mathematical analyses such as Pearson’s correlation coefficients and R-squared values for pairs of elements could give a glimpse of chemical or functional interactions.

Trace element levels in normal and pathological brain development are central to establish cause and effect relationships, mainly for nutritional deficiencies or metal transporter defects and other disorders in which trace elements are involved. Nowadays, many diseases have been regarded as neurodevelopmental disorders, with its roots planted in the first months or years of life. Disturbances in early brain development have been deemed important for later developing Parkinson’s disease or schizophrenia (Piper et al., 2012; Le Grand et al., 2015). Therefore, it seems reasonable to emphasize the need to gather data on normal trace element levels in embryonic brain tissue as presented in this work. In conclusion, cerebral organoids derived from pluripotent stem cells recapitulate features of trace element constitution previously described in the human brain.

Supplemental Information

Figure S1 Characterization of induced pluripotent stem cells

(A) RT-PCR analysis for pluripotency markers. GAPDH was used as loading control. (B) Differentiation of iPS cells into cells derived from endoderm (alpha-fetoprotein, AFP), mesoderm (alpha smooth muscle actin, SMA) and ectoderm (β-tubulin III) in embryoid bodies assay. Scale bars: 100 µm.

Click here for additional data file.

Figure S2 Cerebral organoids derived from human induced pluripotent stem cells

(A) Cerebral organoids of 45-days old produced from iPS cells derived from skin fibroblasts present similar coloring and texture to hESC-derived cerebral organoids. (B and B’) A 30-days old organoid in detail showing different hues according to different cell layers. (C) Along differentiation, organoids’ diameter doubled between days 7 and 30 in culture and tripled after 45 days. (D) Organoid section stained for β-tubulin III and PAX6 to show distintic neuronal and neural progenitor cells distribution, respectively. (E) DAPI stained nuclei. (F) Merged channels. The graph represents mean ±S.D. n = 8 for 7-days old organoids, n = 15 for 15-days old organoids, n = 7 for 30-days old organoids, n = 10 for 45-days old organoids. P < 0.05 for 7-days old versus 30 and 45-days old organoids, for 15-days old organoids versus 30 and 45-days old organoids, and for 30-days old organoids versus 45-days old organoids. Cerebral organoids were obtained from one assay. Scale bars: A = 1.5 mm, B = 250 µm, B’ = 75 µm, D–F = 100 µm.

Click here for additional data file.

Table S1 Cell culture media background signals in SR-XRF analyses

Cerebral organoid differentiation media were analyzed by SR-XRF in order to assess background signals generated by elements present in culture media (EB medium, neuroinduction medium and neurodifferentiation medium). Values are shown as percentage of total values found in cerebral organoids. Elements Fe and Zn were not detected (ND) within XRF range.

Click here for additional data file.

Table S2 Matrigel background signals in SR-XRF analyses

Matrigel was used during organoid formation; therefore, it was analyzed by SR-XRF in order to assess background signals generated by this reagent. Values are shown as percentage of total values found in organoids. Elements Fe and Zn were not detected (ND) within XRF range.

Click here for additional data file.

Table S3 Reports on cerebral cortex elemental concentration

Values from Rajan et al. (1997) are an average from different regions of cerebrum cortex and S.D. refers to the number of different regions assessed.

Click here for additional data file.

Data S1 Diameter size of cerebral organoids derived from ES cells along differentiation

Click here for additional data file.

Data S2 Statistical analysis of the relative area occupied by ventricles comparing 30-days old versus 45-days old cerebral organoids

Click here for additional data file.

Data S3 Statistical analysis of the number of PH3 positive cells/mm2 comparing 30-days old versus 45-days old cerebral organoids

Click here for additional data file.

Data S4 Statistical analysis of the positive area for MAP2 staining comparing 30-days old versus 45-days old cerebral organoids

Click here for additional data file.

Data S5 Statistical analysis of the number of GAD67 positive cells/mm2 comparing 30-days old versus 45-days old cerebral organoids

Click here for additional data file.

Data S6 Statistical analysis of glutamate fluorescence intensity in the cerebral organoids’ border comparing 30-days old versus 45-days old cerebral organoids

The fluorescence intensity in the cerebral organoids’ border was normalized for the tissue background and was given as fold increase on basal condition.

Click here for additional data file.

Data S7 Diameter size of cerebral organoids derived from iPS cells along differentiation

Click here for additional data file.

Data S8 Statistical analysis of cell density comparing organoid center and organoid edge in cerebral organoids

Click here for additional data file.

We are indebted to Ismael Gomes, Gabriela Lopes Vitória, Marcelo Costa and Jarek Sochacki for taking excellent care of cell lines.

Additional Information and Declarations

Competing Interests

Author Contributions

Human Ethics

Data Availability

The authors declare there are no competing interests.

Rafaela C. Sartore conceived and designed the experiments, performed the experiments, analyzed the data, wrote the paper, prepared figures and/or tables, reviewed drafts of the paper.

Simone C. Cardoso conceived and designed the experiments, performed the experiments, analyzed the data, contributed reagents/materials/analysis tools, wrote the paper, prepared figures and/or tables, reviewed drafts of the paper.

Yury V.M. Lages performed the experiments, reviewed drafts of the paper.

Julia M. Paraguassu performed the experiments.

Mariana P. Stelling performed the experiments, analyzed the data, prepared figures and/or tables, reviewed drafts of the paper.

Rodrigo F. Madeiro da Costa performed the experiments, analyzed the data, reviewed drafts of the paper.

Marilia Z. Guimaraes conceived and designed the experiments, analyzed the data, wrote the paper, prepared figures and/or tables, reviewed drafts of the paper.

Carlos A. Pérez analyzed the data, contributed reagents/materials/analysis tools, reviewed drafts of the paper.

Stevens K. Rehen conceived and designed the experiments, analyzed the data, contributed reagents/materials/analysis tools, wrote the paper, reviewed drafts of the paper.

The following information was supplied relating to ethical approvals (i.e., approving body and any reference numbers):

Comitê de Ética em Pesquisa do Hospital das Clínicas da Faculdade de Medicina (CAPPesq, HCPA) (IRB00000921)

Comitê de Ética em Pesquisa do Hospital Copa D’Or (CEPCOPADOR) (727.269).

The following information was supplied regarding data availability:

The raw data has been supplied as a Supplementary File.

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
