# Peer review of "Trace elements during primordial plexiform network formation in human cerebral organoids"

_PeerJ, doi:10.7717/peerj.2927_

## Round 0.1 · original submission · Major Revisions

The comments of both reviewers reflect the need for clarification and careful editing of the revised manuscript. Any revisions must also carefully address the methodological and statistical concerns raised by Reviewer 2 before it can be considered for publication. Please note that re-review will be required prior to any favourable decision. I appreciate the time and effort required to address these concerns, and I look forward to reading a revised submission.

·

Basic reporting

This paper is generally very well written and easy to follow.

Some minor corrections need to be made:
Line 79: The end of the sentence beginning "While great scientific..." doesn't make sense.
Line 88: The abbreviation SR-XRF doesn't need proper nouns (capitalised).
Line 119: "However, differently from the latter study..." should read "In these experiments, most of the protocol..." or similar.
Line 196: Please reference PyMca (V.A. Solé, E. Papillon, M. Cotte, Ph. Walter, J. Susini, A multiplatform code for the analysis of energy-dispersive X-ray fluorescence spectra, Spectrochim. Acta Part B 62 (2007) 63-68.)
Line 292: Element names do not need to use capital letters.
Line 310 onwards: Why are concentrations in parentheses?
Discussion: From the discussion, element symbols have been replaced with element names. Please make sure use is consistent throughout.
Line 466: Brain regions are not proper nouns.
Fig 6: Shouldn't 'Elemental distribution' be preferable to 'Atomic distribution'?
Fig 6: Scales are not labelled correctly, what concentration do they refer to? They are also difficult to read, and use of multiple decimal places could be fixed.

Experimental design

It is not clear how quantitative information was obtained. Please expand on the methods used to generate concentration data for the images. What standards were used? I assume the levels are per gram of material?

Validity of the findings

We have previously reported that cultured neurons and astrocytes are massively deficient in iron compared to in vivo models (Hare et al, Metallomics, 2013, 5, p.1656-1662). When discussing comparisons (Table 2), would the authors care to comment on this?

I agree that commenting on sources of association is speculative, but I would not be opposed to more - can the authors comment further on potential reasons for colocalisation of specific metals? It is important to note that metal levels alone provide limited information, and identifying associated biomolecules is important. How would the authors potentially address such studies in the future?

Additional comments

I enjoyed this paper and look forward to its publication after minor changes.

Reviewer 2 ·

Basic reporting

(minor) There are a number of typographical errors or unclear phrasings in the text. These did not affect my understanding of the science and can easily be addressed by a scientific language editor.

Experimental design

1) In Figure 2 and in line 154 of the text, the authors appear to consider any cavity with dense cellularity around it to be a ventricle. This seems a very loose definition of what a ventricle may be. Such a formation could also be a non-ventricle structure such as rosette or pseudorosette, or possibly even an air bubble in the matrigel which stimulated cells to cluster nearby. This is especially a concern for the 45-day organoids (Fig 2B) where there are several small lumen-containing structures without the clear cell layering present in the 30-day organoid (Fig 2A). The immunofluorescence staining in Figure 3 is very helpful to demonstrate that some of these structures in the 30-day organoids have ventricle-like properties. However such staining has not been performed for the 45-day organoids. Similar staining of ventricle or SVZ markers (such as PAX6 and TBR2 as in Figure 3, or other markers appropriate for this developmental stage) around ventricles in the 45-day organoids (Figure 4) would help alleviate this concern. The opinion of a neuropathologist on these ventricle-like structures would be appreciated as well.

2) Figure 1D demonstrates that 45-day organoids are appreciably larger (nearly twice the diameter) of the 30-day organoids. However in Figure 2 panels A and B, the representative 45-day organoid is similar in size (in fact smaller) than the 30-day organoid, suggesting that these may not be very representative images. The data in Figure 2C is also difficult to interpret because ventricle size is normalized to total organoid area. Since the average 45-day organoid is nearly twice as big as a 30 day organoid (Fig. 1D), if we assume the same average number of ventricles per organoid, the data in 2C suggests that the 45-day ventricles are about the same size as the 30-day ventricles, not smaller as the authors state. Another possibility is that there are less ventricles per 45-day organoid, regardless of size. I suggest a slightly different approach: for each time point, the authors could measure the total area of each ventricle and graph this (average +/- std.dev. in µm2), and also include the average number of ventricles counted per organoid.

3) Figure 4A, line 271 – the text states that MAP2 staining identifies the NPP layer at the organoid surface, but the imaging data and figure legend do not support this and MAP2 staining appears to be positive everywhere.

4) Figures 3, 4, and 5 are highly connected, rely upon comparisons between each other (even though the image data are not shown for each time point), and build to the same message: the 30-day organoids are highly proliferative and the 45-day organoids are more differentiated and less proliferative. As an optional suggestion for the authors: I feel that the paper would be easier to grasp and flow better if these data were restructured into 1 figure. Perhaps 3 columns (30-day image, 45-day image, graph) and 4 rows (PH3, MAP2, GAD67, Glutamate). This way readers could easily see and interpret the differences between the two time points at a glance.

5) The SR-XRF data and element distribution is the most important and novel data in this work, but is lacking some key controls. The authors found that P, S, K, Fe, Ca, and Zn were present in the organoids. Is this a surprise, or would the authors expect these to be the major trace elements in any tissue sample? Is this signature more similar to brain tissue compared to most other tissues, or is it unique to organoid culture? How much of this signal is due to the presence of these elements in the matrigel that forms the organoids? The authors should use matrigel without cells, formed into organoids, and cultured in the same conditions as the real cell-containing organoids as controls for these studies. Ideally, comparison to brain and other tissue SR-XRF data would be valuable here.

6) The number of independent organoids used and the number of slices from each organoid in the SR-XRF and trace element distribution experiments should be indicated. I could not find this information in the paper.

7) As suggested above, figure 6 needs Matrigel-only organoids (no cells) as a control to understand the element contribution from the forming tissues vs. the background. Also please be aware that matrigel is biologically produced and has high lot-to-lot variation. Therefore the same lot of matrigel (preferably at the same time) must be used for the background organoids, otherwise the data could have strange variations that may be difficult to compare.

8) The color scales in figure 6 are slightly confusing. The 45-day Fe color scale range appears to be dominated by a single very bright red pixel, leaving the tissue blue, apparently near zero. But this is not accurate since nearly all the dynamic range is compressed into the bottom ~10% of the scale. It also makes comparison to the 30-day sample difficult because the top of this scale is an order of magnitude higher than the 30-day Fe scale. Similarly, the 30-day Ca scale also makes the entire tissue appear near zero, but the top of the scale is an order of magnitude higher than the 45-day Ca scale. The authors could simply use identical scales between the time points of each element as they have done for Sulfur (S).

9) Several elements are at higher concentration near the edges but nothing appears higher in the center of the organoids. The gradients seem either radial or uniform but I do not see an indication of different gradients around ventricle structures (or any cellular structures) at either timepoint. This raises the question of how these element concentrations relate the structures described earlier in the paper. Are there ventricles present as expected from earlier figures? Where are they and do they match any elemental variations? Are the presence of some of the elements that are increased around the edges simply an effect of cell density, reflecting denser cells near the organoid edges?

10) The authors present estimates of trace element concentration in Table 2. There are some problems here. Again we need to know the number of individual organoids that were tested to produce this data. The authors claim that “In general, the levels of trace elements tend to diminish from 30 to 45 days of differentiation.” The authors cannot make this claim in general, or even for most elements tested. Only P and K have a clear change (need p-values), whereas the other elements tested have no change beyond the standard deviations and/or trend in the opposite direction. I believe that this claim should be removed from the text if it cannot be supported. The authors present the manufacturers’ listed concentrations of trace elements in the media used. This is not sufficient to control for background. The media’s Calcium concentration almost completely accounts for the calcium levels found in the organoids. As noted above, the authors again neglect matrigel-only organoids (without cells) as a critical control in these experiments. For ease of understanding and clarity I suggest that this data be presented graphically. This could be combined into figure 6 if the authors choose to do so.

11) The authors demonstrate a positive correlation spatially between everything that they tested. This could indicate that the relationship between trace elements studied here mostly reflects a technical/modeling effect such as cell density rather than any biological effect. For instance, areas of organoids with many cells are likely rich in all the trace elements. The authors should test for this since it would greatly influence the interpretation of the results. Does the higher correlation in the 45-day organoids indicate a more spatially defined cell distribution as might be expected for more developed tissue structures?

Validity of the findings

As noted above, the number of independent organoids used and the number of slices from each organoid in the SR-XRF and trace element distribution experiments should be indicated. I could not find this information in the paper. p-values should be indicated. The inclusion of additional experimental controls and perhaps additional organoid replicates as noted above will allow a judgement of the validity of these findings.

Additional comments

Sartore et. al. present a study characterizing the inclusion of trace elements during two modeled stages of brain development. To achieve this, the authors have utilized cerebral organoids as a surrogate for the developmental brain. This is a good use of newly developed organoid technologies since it allows the authors to ask a question using human tissue that cannot be explored using either clinical samples or mouse models. The recapitulation and validation of cerebral organoid structures is incomplete but what is shown is also convincing. My greatest concerns with this work are within the elemental measurement sections, specifically the lack of acellular organoid controls and the need to account for cell density. Without this additional data it is difficult to judge whether the data presented are due to technical artifact of the organoid models used. With a number of improvements in several places, this work could be a potential candidate for future publication.

---

## Round 0.2 · accepted · Accept

Both reviewers indicated that the revised manuscript is substantially improved, and I believe it will make an excellent addition to the PeerJ library.

·

Basic reporting

n/a

Experimental design

n/a

Validity of the findings

n/a

Additional comments

I am satisfied that all issues raised have been addressed.

Reviewer 2 ·

Basic reporting

I am glad that the authors found the criticisms of both reviewers helpful and were responsive to our concerns.

Experimental design

I am glad the authors agreed with many of my suggestions and I feel the clarity in this version of the manuscript is increased.

Validity of the findings

The recent inclusion of background control data and the demonstration any elemental background is not significant in these experiments have addressed my previous major concern with these studies. I would have been nice for a minimum of 3 separate specimens to be quantified, however I do understand the technical, financial, and analytical challenges with regard to repetition of these particular experiments.

Additional comments

I feel that the authors have clearly improved their manuscript, I thank them for their responsiveness to my many criticisms.